# Photoplethysmography Analysis with Duffing–Holmes Self-Synchronization Dynamic Errors and 1D CNN-Based Classifier for Upper Extremity Vascular Disease Screening

**Pi-Yun Chen [1], Zheng-Lin Sun [1], Jian-Xing Wu [1], Ching-Chou Pai [2], Chien-Ming Li [3], Chia-Hung Lin [1,\*]** and **Neng-Sheng Pai [1,\*]**

1  Department of Electrical Engineering, National Chin-Yi University of Technology,
   Taichung City 41170, Taiwan; chenby@ncut.edu.tw (P.-Y.C.); eechl53@ncut.edu.tw (Z.-L.S.);
   jian0218@gmail.com (J.-X.W.)
2  Show-Chwan Memorial Hospital, Division of Cardiovascular Surgery, Changhua 50091, Taiwan;
   actirin1945@gmail.com
3  Chi Mei Medical Center, Department of Medicine, Division of Infectious Diseases, Tainan City 41170, Taiwan;
   235813cmli@gmail.com
\*  Correspondence: eechl53@gmail.com (C.-H.L.); pai@ncut.edu.tw (N.-S.P.)

**Abstract:** Common upper limb peripheral artery diseases (PADs) are atherosclerosis, embolic diseases, and systemic diseases, which are often asymptomatic, and the narrowed arteries (stenosis) will gradually reduce blood flow in the right or left upper limbs. Upper extremity vascular disease (UEVD) and atherosclerosis are high-risk PADs for patients with Type 2 diabetes or with both diabetes and end-stage renal disease. For early UEVD detection, a fingertip-based, toe-based, or wrist-based photoplethysmography (PPG) tool is a simple and noninvasive measurement system for vital sign monitoring and healthcare applications. Based on time-domain PPG analysis, a Duffing–Holmes system with a master system and a slave system is used to extract self-synchronization dynamic errors, which can track the differences in PPG morphology (in amplitudes (systolic peak) and time delay (systolic peak to diastolic peak)) between healthy subjects and PAD patients. In the preliminary analysis, the self-synchronization dynamic errors can be used to evaluate risk levels based on the reflection index (RI), which includes normal condition, lower PAD, and higher PAD. Then, a one-dimensional convolutional neural network is established as a multilayer classifier for automatic UEVD screening. The experimental results indicated that the self-synchronization dynamic errors have a positive correlation with the RI ($R^2 = 0.6694$). The *K*-fold cross-validation is used to verify the performance of the proposed classifier with recall (%), precision (%), accuracy (%), and $F_1$ score.

**Keywords:** upper extremity vascular diseases; wrist-based photoplethysmography; Duffing–Holmes system; 1D convolutional neural network

## 1. Introduction

In Taiwan, patients with Type 2 diabetes or with both diabetes and end-stage renal disease have an increased risk factor for atherosclerotic cardiovascular disease (CVD) [1,2]. Approximately two-thirds of those patients died of related CVDs, which are the usual causes of heart attacks, strokes, high blood pressure, and peripheral artery disease (PAD) [3,4]. PAD is a progressive disorder disease and is caused by stenosis of large-sized or medium-sized arteries. More than 200 million people have PAD [5]. Its prevalence increases substantially with age and has a higher risk in the elderly population. Atherosclerosis has a more than 90% prevalence in PAD, and more than 50% of PAD patients are asymptomatic. PAD's symptoms are related to the arteries in the upper or lower limbs, including discomfort or pain in the arms or legs, cramping or weakness in one or both upper arms, diabetic foot, and upper or lower extremity vascular diseases (UEVD or LEVD). UEVDs are caused by atherosclerosis, embolic disease, or systemic diseases and anatomic abnormalities, such

as aortitis and thoracic outlet syndrome. For medical exams, diagnostic methods, such as digital angiography, magnetic resonance angiography, or computed tomography angiography, can be performed to detect UEVD or LEVD [3,4,6,7]. However, the aforementioned diagnostic methods are not readily available for in-home healthcare or nonmedical environment. Mobile or portable sensors can be embedded in a watch or a wristband with a mobile phone display, such as smartphones and smartwatches, to measure biosignals [8,9]. These mobile measurement devices are applied to mobile and personalized health management and can measure and transmit symptom biosignals via a wired or wireless communication network for telecare applications.

Among these portable sensors, photoplethysmography (PPG) is an optical technique to obtain a plethysmogram for noninvasively detecting blood oxygen saturation (SpO$_2$) and blood volume changes for further time-domain and frequency-domain analyses. As shown in Figure 1, a portable PPG measurement tool, such as fingertip-based, toe-based, or wrist-based PPG, can provide nonclinical and noninvasive continuous health monitoring and can be applied to severe acute respiratory syndrome (COVID-19) and chronic disease symptom detection and monitoring [10,11], including heart rate (HR), heart rate variability (HRV), SpO$_2$, and respiratory rate (RR). Moreover, PPG waveform features and shapes with blood volume changes in the time domain are used to evaluate arterial stiffness and compliance, such as pulse interval for HR analysis, stiffness index (SI) for large artery stiffness analysis, and reflection index (RI) for small to medium-sized arterial stiffness analysis. For SI and RI extraction, the systolic peak, dicrotic notch, and diastolic peak in each PPG pulse need to be obtained for ratio index computations [8,12–14]. The systolic peak indicates pulsatile changes in blood volume and is proportional to local vascular distensibility. Meanwhile, the dicrotic notch and diastolic peaks often disappear, and the time delay between systolic and diastolic peaks decreases with age and chronic diseases [15]. Hence, the evaluation of arterial stiffness in the time-domain pulse waveform is difficult. The first derivative PPG (i.e., velocity waveform of PPG signals) is needed to locate the systolic peak, dicrotic notch, and diastolic peak in each PPG signal [8,15].

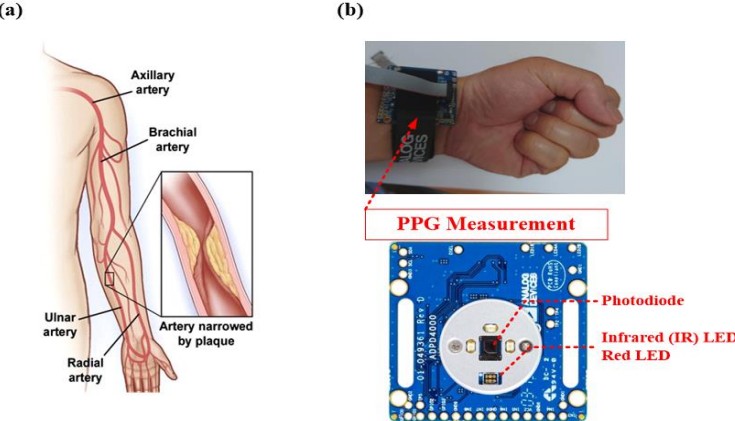

**Figure 1.** Wrist-based photoplethysmography (PPG) measurement tool for detecting UEVD. (**a**) Narrowed arteries (stenosis) in the upper limb, (**b**) Wrist-based PPG measurement tool.

In frequency-domain analysis, HRV spectral analysis with fast Fourier transform (FFT) provides information about autonomic nervous system activity, such as low-frequency components for sympathetic and parasympathetic activities and high-frequency components for parasympathetic activity [16–18]. Moreover, the frequency spectrum of PPG signals can use the main characteristic frequency to estimate the RR and indicate that PPG signals have the characteristics of high-frequency components in healthy subjects and low-frequency components in PAD patients. However, PPG signals may contain various sources of noises, such as the subject's motion and respiration, which are required to remove the slow nonstationary trends or unwanted artifacts from PPG signals. Hence, before HRV spectral

analysis, a detrending method or a high-pass or band-pass filter (i.e., Butterworth filter, within the band limit 0.4–3.5 Hz) is used to reduce the detrimental effects [19–21]. However, frequency spectral parameters cannot screen the PAD risk level. In time-domain analysis, the PPG's transit time and amplitudes are differences between the normal and abnormal PPG signals, which indicates difference increases as the PAD gradually severity and decrease in amplitudes and also indicates that PPG waveform changes relate to synchronous cardiac changes in vascular blood volumes.

Hence, in this study, we propose a time-domain quantizer and a classifier for automatic UEVD screening. In time-domain PPG analysis, synchronizing chaotification methods (SCMs), such as the Duffing–Holmes (D–H), Sprott, or Chen–Lee systems [21–27], have been applied to signal preprocessing and feature extraction. With SCMs, a discrete self-synchronization dynamic system consisting of a master system (MS) and a slave system (SS) can be established as a time-domain quantizer for extracting different features from normal and abnormal PPG signals, as shown in Figure 2. SCMs have advantages in the broad frequency spectrum and fractal properties of motion in the phase space for analyzing PPG changes in amplitude (systolic peak) and time delay (systolic peak to diastolic peak). To simplify the high-dimensional SCMs, the D–H system [21,23,24,28,29] can be linearized, simplified, and reduced as a discrete two-dimensional (2D) self-synchronization dynamic system [21,27]. For digital signal processing feeding two different PPG signals into the MS and SS can cause the chaotic system to produce chaotic phenomena. For feature extraction, normal and abnormal PPG signals are fed into the MS and SS, respectively, so that the D–H system can track the differences. Any changes in the PPG waveform, such as initial conditions, amplitudes, or phases can lead to changes in the dynamic errors, the so-called self-synchronization dynamic errors, and produce different scatter maps of dynamic trajectories and chaotic regions for screening the normal condition (Nor), lower PAD (LPAD), and higher PAD (HPAD). Comprehensive dynamic errors ($\Phi$) have a positive correlation with the RI index (RI index versus $\Phi$ as shown in Figure 2) and will be validated in this study. Hence, a dynamic error-based quantizer can be used to quantify the relationships of the dynamic errors versus PAD risk levels, which provides scaling indications for rapidly screening PAD. Hence, chaotic scatter maps can be applied to classification tasks. For the automatic screening function, a one-dimensional (1D) or 2D convolutional neural network (CNN) [30–35] can be used to design a multilayer classifier consisting of the convolutional layers, pooling layers, flattening layers, and a fully connected network. In previous studies [33–35], PPG analysis with 2D CNN-based classifiers has been applied to HR and hypertension detection, PAD-related CVD risk detection, and PAD detection, exhibiting high accuracy for screening mild to severe diseases, arterial hardening, and Nor. However, a 2D CNN with a deep learning algorithm [33–35] has some drawbacks and limitations [30], that is, the number of 2D CNN multilayers need to be determined, the computational complexity for training 2D CNN is high, the 2D CNN is unsuitable for real-time signal screening applications on a mobile or portable device, and a large dataset is needed to train the 2D CNN. Moreover, the 2D CNN algorithm needs to be performed with a graphics processing unit to accelerate the overall process.

Hence, to improve the drawbacks of 2D CNNs, a 1D CNN with a time-domain feature extraction layer (i.e., D–H-based quantizer), 1D convolution processes, 1D subsampling processes, and a few hidden nodes and layers in the fully connected network is used to design a multilayer classifier for automatic UEVD screening. This adaptive learning model can also be deployed to adjust the optimal network parameter with feeding training patterns using the gradient descent method or particle swarm optimization algorithm [31,36–39], which provides a labeled dataset to maximize classification accuracy. The optimization algorithm with iteration processes is used to refine the optimal network parameter for applying to enhance the classifier performance. In the convolution-subsampling layer, the 1D convolution and subsampling processes are employed to enhance the features, reduce unwanted noises, and protect the vector's dimension from self-synchronization dynamic errors. In the classification layer, the gradient descent algorithm can directly use

the gradient values for finding the optimal network parameter. In addition, an appropriate selecting learning rate can speed up the training processes, and then the convergence speed is increased. The optimized classifier can be applied to separate the Nor from LPAD and HPAD for UEVD classification. Hence, the proposed 1D CNN-based classifier can directly analyze and process raw 1D measured PPG signals and learn to extract features in pattern recognition, resulting in low computational operations for real-time signal screening applications. The proposed screening procedure is tested using the PPG measurement data, which are obtained from 40 subjects (including 11 Nor, 11 LPAD, and 22 HPAD) and approved by the hospital research ethics committee. The measurement data are divided into training and testing datasets to train the classifier and validate the classifier's feasibility. The experimental results indicated the computational efficiency and high accuracy against random noises of the proposed screening procedure using the measurement data for UEVD screening.

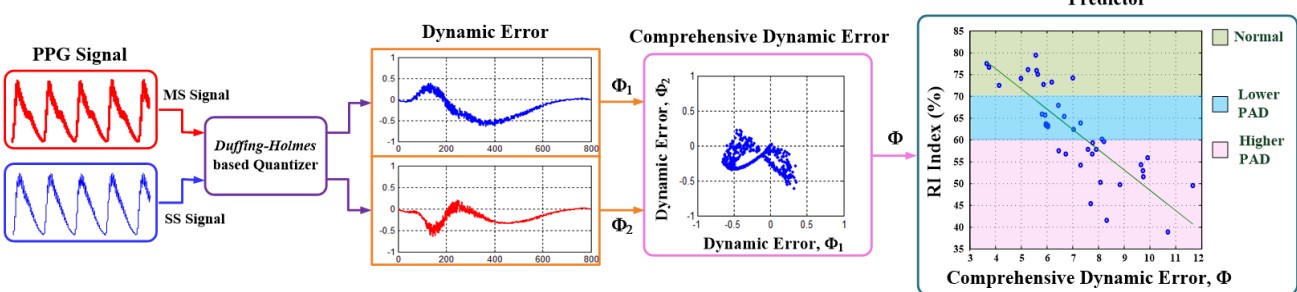

**Figure 2.** Duffing-Holmes (D–H)-based quantizer for feature extraction from normal and abnormal PPG signals.

The remainder of this study is organized as follows: Section 2 describes the methodology, including the PPG measurement, D–H-based quantizer, feasibility assessment, and 1D CNN-based classifier. Sections 3 and 4 present the feasibility tests and results and the conclusions, respectively.

## 2. Materials and Methods

### 2.1. PPG Measurement

As shown in Figure 1b, the PPG prototype board (EVAL-ADPD4100Z, Analog Devices Inc., Wilmington, MA, USA) consists of light sources and a photodetector. The light sources are three green light-emitting diodes (LEDs), one infrared (IR) LED, and one red LED, which are separately driven by the EVAL-ADPDUCZ (Cortex M4) microcontroller board (Analog Devices Inc., Wilmington, MA, USA) [40], and a single 7 mm$^2$ photodiode without optical fiber coating is incorporated for optical measurement. The optical measurement system of the PPG prototype board works in reflection mode. In this study, we select the IR or red LED or both of them to measure PPG signals and SpO$_2$ for medical diagnostics and healthcare applications. This wrist-based PPG wearable tool is applied to clinical physiological monitoring and vascular assessments, such as HRV, HR, SpO$_2$, arterial disease, and arterial compliance or aging monitoring. The microcontroller can repackage the PPG measurement data and send them to a virtual serial port through the USB to a laptop. Meanwhile, the wrist-based PPG tool sits flat on the skin for each PPG measurement from the wrist, the PPG raw data can be displayed in real time with limited latency and subjected to frequency-domain and time-domain analyses by the Wavetool Evaluation Software graphical user interface. Moreover, the raw data stream is connected to external analysis programs, such as LabVIEW® (National Instruments Corp., Austin, TX, USA) or MATLAB® (1994–2021, The MathWorks Inc., Natick, MA, USA) software in real time. Therefore, the measured PPG signal can be preprocessed and analyzed on a laptop. A PPG data stream, $PPG_{org}$, is preprocessed using the detrending process [19–21] and normalized as follows:

$$PPG_{det} = detrend(PPG_{org}) \tag{1}$$

$$PPG = \frac{PPG_{det}}{max(PPG_{det})} \tag{2}$$

where detrend (●) is a detrending operator, which is used to remove unwanted variations and direct current bias from the original measured *PPG* signal; $PPG_{det}$ is the detrended *PPG* signal; *PPG* is the normalized data stream, and *max* (●) is an operator used to find the maximum value in a detrended stream.

## 2.2. Duffing–Holmes-Based Quantizer

Based on a forced oscillator, the second-order D–H differential equation can be expressed as follows [21,23,24,28,29]:

$$\ddot{x} + \delta\dot{x} - x + x^3 = f\sin(\omega t) \tag{3}$$

where $\delta$ is the damping parameter, $f$ is the excitation amplitude, and $\omega$ is the excitation frequency. Equation (3) is a nonlinear dynamic system, that is, the so-called D–H system. The dynamic D–H system with excitation amplitude, $f$, and frequency, $\omega$, can generate dynamic chaotic phenomenon using the external influence term, "$f\sin(\omega t)$." The D–H system can be reduced from a high-order to a one-order nonautonomous equation, as follows:

$$\dot{x} = y \tag{4}$$

$$\dot{y} = -\delta y + x - x^3 + f\sin(\omega t) \tag{5}$$

Let excitation amplitude $f$ = 0 (under unforced condition), $x_1 = x$, and $x_2 = y$, Equations (4) and (5) can be linearized as follows

$$\dot{x}_1 = x_2 \tag{6}$$

$$\dot{x}_2 = (1 - x_1^2)x_1 - \delta x_2 \tag{7}$$

where $x_1$ and $x_2$ are the state variables for the D–H system. Equations (6) and (7) can also be represented in matrix form, as follows:

$$\begin{bmatrix} \dot{x}_1 \\ \dot{x}_2 \end{bmatrix} = \begin{bmatrix} 0 & 1 \\ 1 - x_1^2 & -\delta \end{bmatrix} \begin{bmatrix} x_1 \\ x_2 \end{bmatrix} \tag{8}$$

where, when system parameter $\delta > 0$, the dynamic system is equivalent to a "hard spring," and when system parameter $\delta < 0$, the dynamic system is equivalent to a "soft spring" [23]. The damping parameter, $\delta \in [0,1]$, can observe the chaotic oscillations and control the dynamic trajectories within a specific boundary region. The D–H system can be implemented in a simple analog electrical circuit and a digital computer [21,23,24,28,29].

For screening the normal and abnormal PPG signals obtained from the subjects, as shown in Figure 2, we can establish a self-synchronization dynamic error system, including the MS and SS, with the following construction:

$$\text{MS}: \begin{bmatrix} \dot{x}_{1m} \\ \dot{x}_{2m} \end{bmatrix} = \begin{bmatrix} 0 & 1 \\ 1 - x_{1m}^2 & -\delta \end{bmatrix} \begin{bmatrix} x_{1m} \\ x_{2m} \end{bmatrix} \tag{9}$$

$$\text{SS}: \begin{bmatrix} \dot{x}_{1s} \\ \dot{x}_{2s} \end{bmatrix} = \begin{bmatrix} 0 & 1 \\ 1 - x_{1s}^2 & -\delta \end{bmatrix} \begin{bmatrix} x_{1s} \\ x_{2s} \end{bmatrix} \tag{10}$$

where $x_{1m}$ and $x_{2m}$ are the state variables for the MS and $x_{1s}$ and $x_{2s}$ are the state variables for the SS. Two variables, $x_{1m}$ and $x_{2m}$, are the sampling data of the normal PPG signal (as gold standard) and $x_{1s}$ and $x_{2s}$ are the sampling data of the measurement PPG signal,

respectively. The error variables can be expressed as $e_1 = x_{1m} - x_{1s}$ and $e_2 = x_{2m} - x_{2s}$; thus, the error dynamic system can be expressed as follows:

$$\begin{bmatrix} \dot{e}_1 \\ \dot{e}_2 \end{bmatrix} = \begin{bmatrix} 0 & 1 \\ (1 - x_{1m}^2) - (1 - x_{1s}^2) & -\delta \end{bmatrix} \begin{bmatrix} e_1 \\ e_2 \end{bmatrix} \tag{11}$$

When the MS and the SS receive different PPG signals, the dynamic errors can be generated by the self-synchronization tracking processes between the two systems, as shown in Figure 2. Two sequences of dynamic errors, that is, $e_1$ and $e_2$, produce the chaotic trajectories of different PAD risk levels. The damping parameter, $\delta$, can control the chaotic trajectories. For digital D–H system implementation in computing applications, Equation (11) can be modified as a discrete dynamic error system with two discrete error variables, that is, $e_1$ and $e_2$, and expressed as follows:

$$\begin{bmatrix} \Phi_1[i] \\ \Phi_2[i] \end{bmatrix} = \begin{bmatrix} 0 & 1 \\ (1 - (x_m[i])^2) - (1 - (x_s[i])^2) & -\delta \end{bmatrix} \begin{bmatrix} e_1[i] \\ e_2[i] \end{bmatrix} \tag{12}$$

$$\begin{bmatrix} e_1[i] \\ e_2[i] \end{bmatrix} = \begin{bmatrix} x_m[i] - x_s[i] \\ x_m[i+1] - x_s[i+1] \end{bmatrix} \tag{13}$$

As expressed in Equation (13), the sampled data of normal PPG signals can be expressed as $x_m[i] = x_{1m}[i]$ and $x_m[i + 1] = x_{2m}[i]$ and the sampled data of abnormal PPG signals can be expressed as $x_s[i] = x_{1s}[i]$ and $x_s[i + 1] = x_{2s}[i]$, where $i = 1, 2, 3, \ldots, n - 1$ with $n$ denoting the total number of sampled data in a measured PPG stream. The chaotic variations can be quantized by two discrete error variables, that is, $e_1[i]$ and $e_2[i]$. For the dynamic error scatter diagram, the comprehensive dynamic error, $\Phi$, [21] is used to evaluate the PAD risk levels, as follows:

$$\Phi = \sqrt{\sum_{i=1}^{n-1} ((\Phi_1[i])^2 + (\Phi_2[i])^2)}, \ i = 1, 2, 3, \ldots, n - 1 \tag{14}$$

where the dynamic errors $\Phi_1 \in R^{n-1}$ and $\Phi_2 \in R^{n-1}$. The index, $\Phi$, is a "comprehensive quantizer" for screening PAD risk levels, including Nor, LPAD, and HPAD. According to the different PAD risk levels, the dynamic error scatter diagrams indicate different distribution regions around a circle (i.e., $\Phi_1 = \pm 0.5$, $\Phi_2 = \pm 0.5$), as denoted by the pink dashed lines shown in Figure 3. For example, for six subjects, the normal patterns are in a narrow region around the point of origin, and for the HPAD cases, the abnormal patterns have two obvious chaotic eyes. Hence, these feature patterns can be used to separate abnormal patterns from normal patterns.

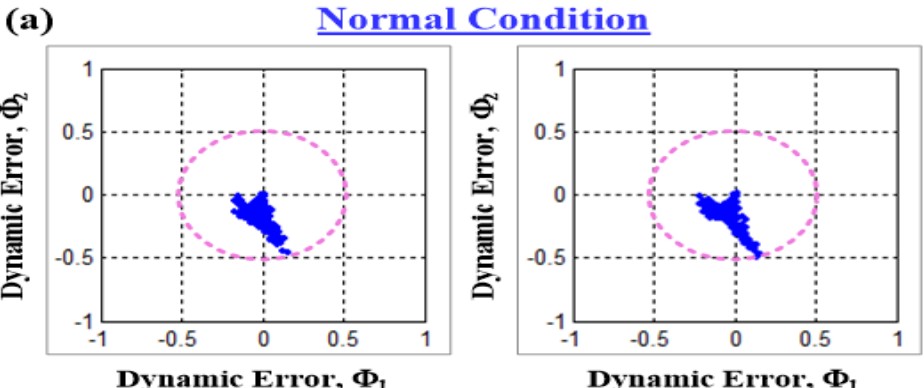

**Figure 3.** *Cont.*

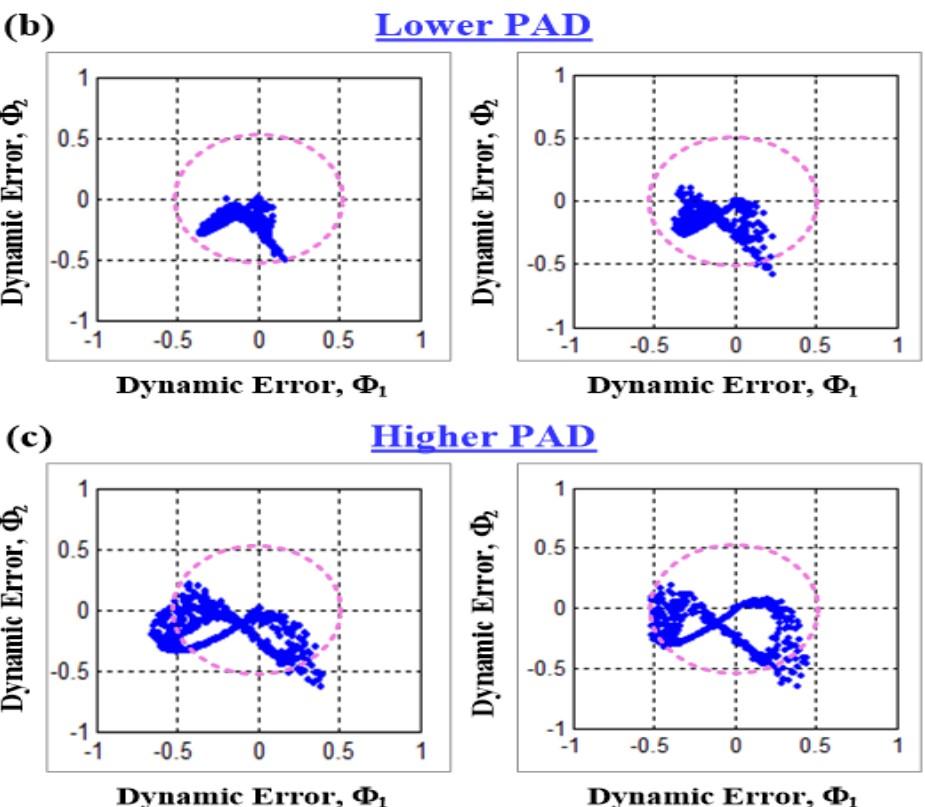

**Figure 3.** Dynamic error scatter diagrams for different risk levels: (**a**) normal condition (Nor), (**b**) lower PAD (LPAD), and (**c**) higher PAD (HPAD).

### 2.3. Time-Domain Analysis with Reflection Index

In the time-domain analysis shown in Figure 4a, a PPG waveform characteristically consists of a systolic peak, dicrotic notch, and diastolic peak. The PPG waveform can be used to assess CVDs and arterial compliance. For arterial compliance assessment, SI and RI [8,12–14] are two arterial stiffness measurement methods: SI is used to measure large arterial stiffness, whereas RI is used to measure small to medium-sized arterial stiffness, such as extreme PADs. Both of them are significantly correlated with age, HR, systolic blood pressure, and diastolic blood pressure [8,12–14]. The RI index can be estimated as follows:

$$RI = \frac{H_2}{H_1} \times 100\% \tag{15}$$

where $H_1$ is the amplitude of the early systolic peak and $H_2$ is the amplitude of the diastolic peak. In this study, the RI is used to assess the stiffness in the extremity arteries of the upper or lower limbs. For 40 subjects (i.e., 11 Nor, 11 LPADs, and 18 HPADs), with the correlation analysis using the quantizer shown in Figure 4b and a linear regression method (i.e., least squares estimation) [21,37], the comprehensive dynamic error, $\Phi$, has a positive correlation with the RI% (RI = $-4.649\ \Phi + 94.939$, $R^2 = 0.6694$). Through preliminary verification, different chaotic feature maps can be used to extract the comprehensive error scales to predict the extreme arterial stiffness levels. In the subsequent section, with the chaotic feature map-based training patterns, the 1D CNN will be established as a classifier for automatically screening the PAD risk levels.

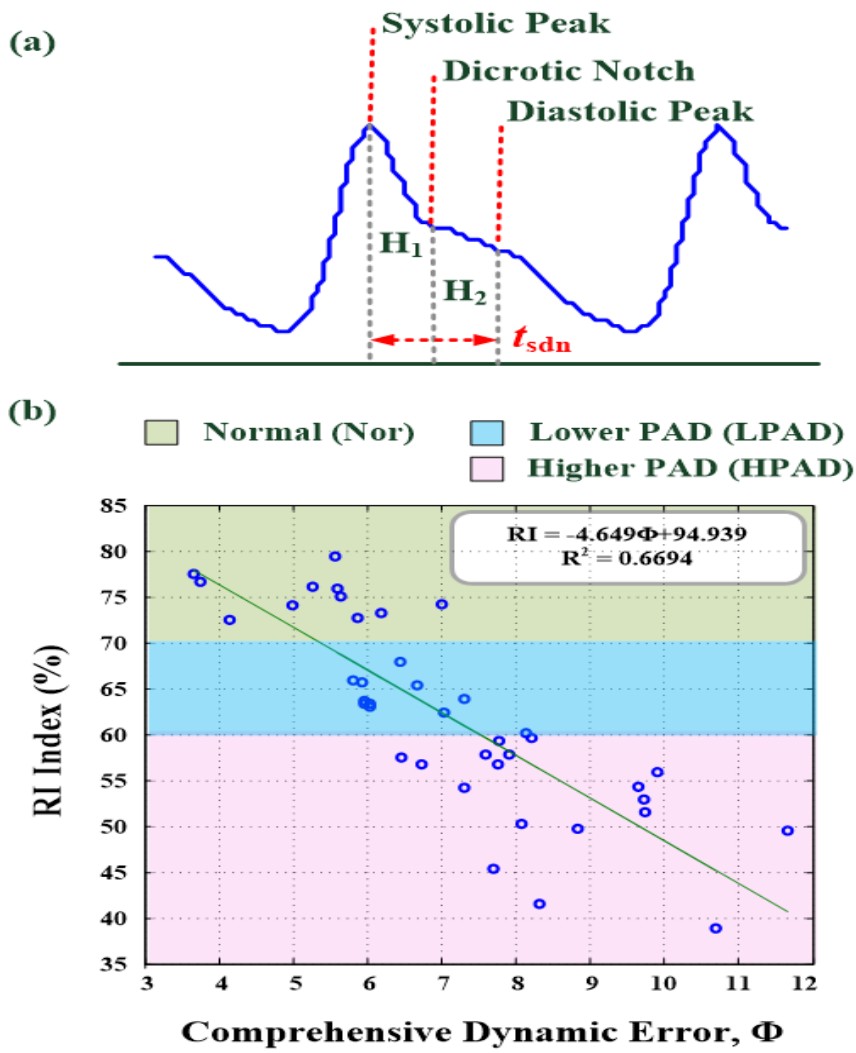

**Figure 4.** (**a**) One PPG waveform consisting of a systolic peak (systole region), dicrotic notch, and diastolic peak. (**b**) Linear regression predictor ($R^2 = 0.6994$) for the reflection index and comprehensive dynamic error, $\Phi$.

### 2.4. 1D CNN-Based Classifier

The conventional CNN is designed to process 2D images or videos exclusively. The 1D CNN is a modified scheme and is preferable to the 2D CNN in dealing with 1D signals (e.g., ECG signals, blood pressure waveforms, vibration signals, and phonoangiographic signals [30–32]) because the computational requirement and complexity of the 1D CNN are lower than those of a 2D CNN, the scheme of the 1D CNN is easier to train and implement that that of a 2D CNN, and training the 1D CNN with a few hidden layers and nodes is faster than training a 2D CNN [30,31]. The configuration of the 1D CNN consists of 1D convolution operations, subsampling (pooling) processes, and a multilayer classifier, as shown in Figure 5. The procedure of the 1D CNN algorithm can be summarized as follows:

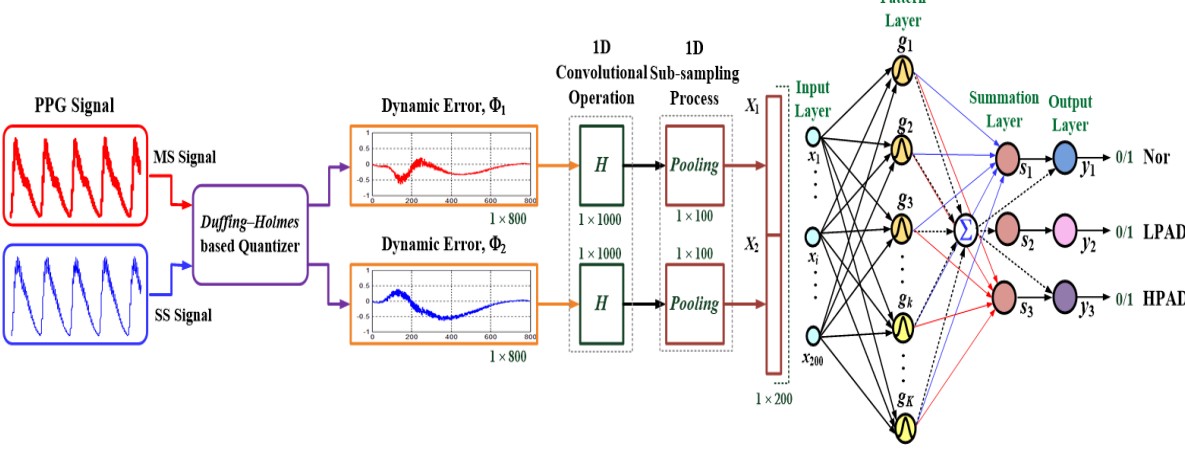

**Figure 5.** Configuration of the proposed 1D CNN-based classifier consisting of the D–H-based quantizer, 1D convolution operations, 1D subsampling processes, and a multilayer classifier.

- 1D convolution operations:

To obtain the feeding dynamic errors, $\Phi_1$ and $\Phi_2$, from the D–H-based quantizer, two 1D digital convolution operators are used to extract the specific characteristics of the dynamic error signals, which can slide over the input signal, $\Phi_e$, $e = 1, 2$, and convolve with it as a feature extraction filter. The convolution calculation $X_e[i] = \Phi_e[i] * H[j]$ (where * is the convolution operator) [31,41,42] is performed using discrete-time convolution, as follows:

$$X_e[i] = \sum_{j=0}^{M-1} H[j]\Phi_e[i - j] \tag{16}$$

$$H[j] = \exp[\frac{-1}{2}(\frac{j-1}{\sigma})^2] \tag{17}$$

where $X_e[i]$ is the finite 1D convolution summation with $i = 0, 1, 2, \ldots, n - 1$, which is an $n + M - 1$ point signal running from 0 to $n + M - 2$; $H[j]$ is a 1D convolution mask used to extract the features of the dynamic error signal, $\Phi_e$ with $j = 0, 1, 2, \ldots, M - 1$ ($M$ is the data length of the 1D convolution mask), which can cover the systole region of each PPG signal; $H[j]$ is the discrete Z-sigmoidal function, and $\sigma$ is the standard deviation. In this study, mask (kernel) size $M = 200$ and sliding stride = 1 are selected. As $j$ runs through 0 to $M - 1$, $X_e[i]$ can be obtained by performing the finite summation of all of the multiplications of the signal, $\Phi_e[i - j]$, and the weighted values of $H[j]$.

After the 1D convolution processes, Equations (16) and (17) can be used to extract the obvious features for classification applications.

- 1D subsampling (pooling) process:

In the pooling process, a 1D downsampling process is used to reduce the vector dimension of the feature pattern and expressed as follows:

$$x_e[i] = X_e[10i], \ I = 1, 2, 3, \ldots, n \tag{18}$$

$$n' = \frac{n + M - 2}{10} \tag{19}$$

where $X_e[i]$, $e = 1, 2$, is the call subsampling feature obtained with the sliding stride = 10 from the feature pattern, $X$. Hence, the vector dimension of the feature pattern can be reduced from $n + M - 2$ to $n'$ ($n' \approx 100$). The pooling process can retain key features and reduce the computational requirement and complexity. Then, two feature patterns are combined as an input pattern, that is, $X = [x_1[i] | x_2[i]] = [x_1, x_2, x_3, \ldots, x_{200}]$, and fed to a multilayer classifier for further screening of the PAD risk levels.

- Multilayer classifier with a fully connected network:

In the classification layer, a fully connected network consisting of an input layer, a pattern layer, a summation layer, and an output layer [31,38–40] is utilized to map the relationship between input feature patterns and PAD risk levels to separate abnormal patterns from normal patterns. For $K$ input–output paired training patterns, with the input pattern ($1 \times 200$ in vector) $X(k) = [x_1(k), x_2(k), x_3(k), \dots, x_{200}(k)]$ and output pattern $Y(k) = [y_1(k), y_2(k), y_3(k)] = $ [Nor, LPAD, HPAD], $k = 1, 2, 3, \dots, K$, these patterns are used to configure the multilayer classifier, including 200 input nodes in the input layer, $K$ pattern nodes in the pattern layer, 4 summation nodes in the summation layer, and 3 output nodes in the output layer, where the output $Y(k)$ denotes one of three risk levels, encoded as (1) Nor: [1,0,0], (2) LPAD: [0,1,0], and (3) HPAD: [0,0,1], with a value of 1 for "possible level" and all other levels encoded with the value of 0. The algorithm of the multilayer classifier is summarized as follows [31,38]:

Step (1) For $200 \times K$ input training patterns, $X(k)$, $k = 1, 2, 3, \dots, K$, is used to set the connecting weights between input and pattern layers.

Step (2) For $K \times 4$ output training patterns, $Y(k) = [y_1(k), y_2(k), y_3(k), 1]$ is used to set the connecting weights between the pattern and summation layers.

Step (3) Compute the outputs of the pattern node, $g_k$, using the radial basis functions (Gaussian functions), as follows:

$$g_k = \exp[-\sum_{k=1}^{K} \frac{(x_i - x_i(k))^2}{2\sigma_k^2}] \tag{20}$$

where $X(0) = [x_1, x_2, x_3, \dots, x_{200}]$ is the testing pattern and $\sigma = \sigma_1 = \dots = \sigma_k$ are the smoothing parameters. The optimal parameter, $\sigma_{opt}$, can be tuned using optimization algorithms, such as the gradient descent method, as follows [38]:

$$\nabla = (T_j(k) - y_j k)(\frac{\frac{\partial s_j(k)}{\partial \sigma} - y_j(k)\frac{\partial g_k}{\partial \sigma}}{g_k}) \tag{21}$$

$$\frac{\partial s_j(k)}{\partial \sigma} = 2(\sum_{k=1}^{K} x_i(k)g_k)(\frac{(x_i - x_i(k))^2}{2\sigma^3}) \tag{22}$$

$$\frac{\partial g_k}{\partial \sigma} = 2(\sum_{k=1}^{K} g_k)(\frac{(x_i - x_i(k))^2}{2\sigma^3}) \tag{23}$$

Hence, the optimal parameter, $\sigma$, can be refined using the iteration computation, as follows:

$$\sigma(p+1) = \sigma(p) + \eta(-\nabla) \tag{24}$$

where $T_j(k)$ is the desired target, $j = 1, 2, 3$; $\eta$ is the learning rate, $0 < \eta \leq 1$, and $p$ is the number of iteration computations. The optimal parameter, $\sigma_{opt}$, can minimize the term "$(T_j(k) - y_j(k))$" in Equation (21).

Step (4) Compute the output $y_j$ in the output layer as follows:

$$s_j = \sum_{k=1}^{K} y_j(k)g_k / \sum_{k=1}^{K} g_k, \, j = 1, 2, 3 \tag{25}$$

$$y_j = \begin{cases} 1, s_j \geq 0.5 \\ 0, s_j < 0.5 \end{cases} \tag{26}$$

where the final outputs, $y_j$, are the binary values representing the three risk levels, that is, Nor, LPAD, and HPAD.

## 3. Experimental Results and Discussion

To validate the proposed 1D CNN-based classifier, the PPG measurement data divided into the training and testing datasets were used to validate its classification feasibility for screening PAD risk levels. A total of 40 subjects were enrolled in a practical investigation, which includes 3 levels, that is, (1) Nor = 11 subjects; (2) LPAD = 11 subjects, and (3) HPAD = 18 subjects. With at least 10 cycles of PPG data (approximately 8 s), the PPG waveforms were different in the time domain for the three risk levels. As shown in Figure 4b, the preliminary screening results indicated that the comprehensive dynamic error, $\Phi$, and RI% were positively correlated ($R^2 = 0.6694$), as shown in Table 1. Hence, the D–H-based quantizer was used to extract features from the PPG stream data directly. As shown in Figure 6, the self-synchronization dynamic errors ($\Phi_1$ and $\Phi_2$) can be applied to the three risk levels identified. The proposed algorithm for the 1D CNN-based classifier is implemented on a tablet PC using a high-level graphical programming language in the LabVIEW and MATLAB software (NI™, Austin, TX, USA). Table 2 shows the related data of the 1D CNN-based classifier, including its layer functions, manners, and feature patterns. The feasibility study was validated as described in detail in the subsequent sections.

**Table 1.** Preliminary results for screening normal, LPAD, and HPAD.

| Pathology Class | $\Phi$ | *RI* (%) |
|---|---|---|
| Normal (11) | $5.34 \pm 0.84$ | $75.20 \pm 4.17$ |
| LPAD (11) | $6.48 \pm 0.83$ | $64.07 \pm 3.91$ |
| HPAD (18) | $8.56 \pm 0.81$ | $52.77 \pm 19.64$ |

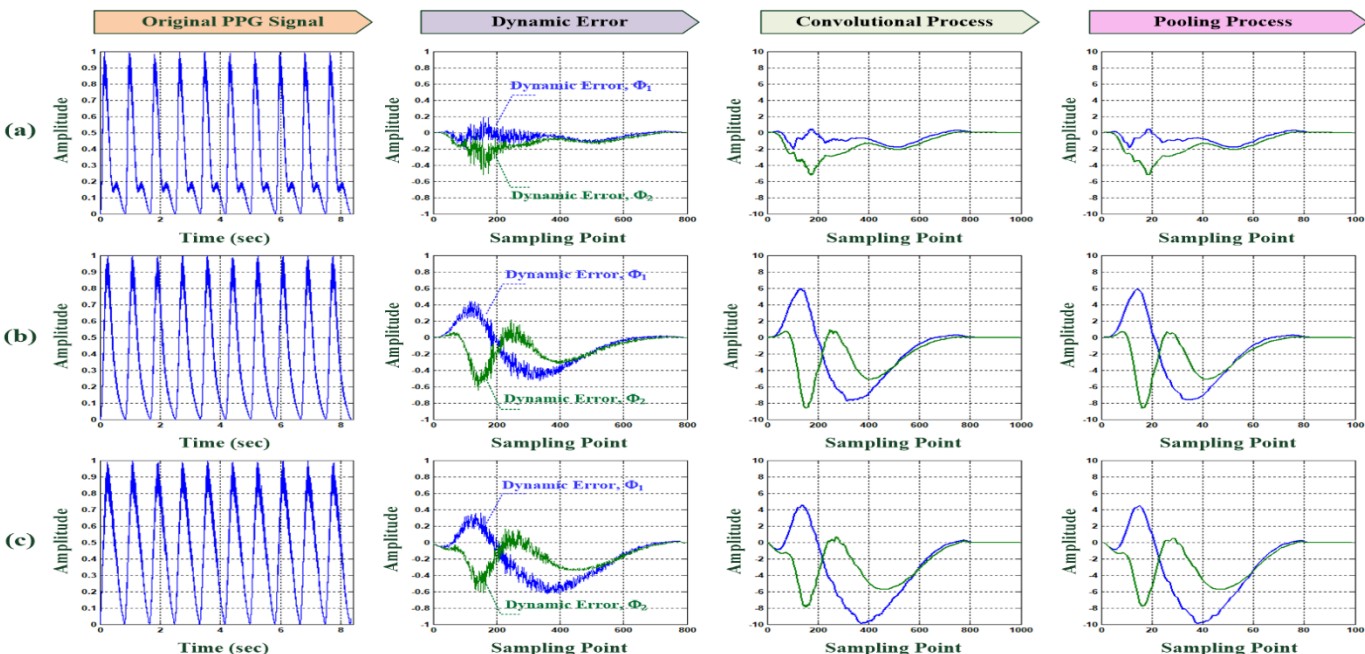

**Figure 6.** Original PPG stream data (10 cycles), self-synchronization dynamic errors ($\Phi_1$ and $\Phi_2$) for each cycle feature extraction, convolution-processed patterns for each cycle feature enhancement, pooling-processed patterns for reducing the pattern's dimension for different PAD risk levels. (**a**) Nor, (**b**) LPAD case, and (**c**) HPAD case.

**Table 2.** Related data of 1D CNN-based multilayer classifier.

| Layer Function | Manner | Feature Pattern |
|---|---|---|
| Feature Extraction Layer | D–H based Quantizer | $\Phi_1$ and $\Phi_2$ ($1 \times 799$) |
| Feature Enhancement Layer | 2 1D Convolutional Operations [40] (stride = 1) | $X_1$ and $X_2$ ($1 \times 998$) |
| Simplifying Feature Layer | 2 1D Pooling Processes (stride = 10) | $x_1$ and $x_2$ ($1 \times 100$) |
| Classification Layer | Multilayer Classifier: 200 input nodes, 40 pattern nodes, 4 summation nodes, 3 output nodes | Input Pattern of Fully Connecting Network: $[x_1 \mid x_2]$ ($1 \times 200$) |
| | Learning Algorithm: Gradient Descent Method | |

### 3.1. Feasibility Tests Using for PPG Feature Extraction

For 40 subjects, PPG raw data with a data length of approximately 8 s (10 cycle PPG signals, 8000 sampling points) were obtained using the noninvasive optical measurement. As shown in Figure 6, first, the measured PPG raw data were detrended and normalized to remove the slow nonstationary trends and unwanted artifacts using Equations (1) and (2). Then, with the 10-cycle PPG stream data, as shown in Figure 7, the D–H-based quantizer with the damping parameter, $\delta = 0.30$, was employed to track the differences between the normal templated PPG stream data and the incoming (unknown) PPG stream data and generate the self-synchronization dynamic errors, $\Phi_1$ and $\Phi_2$. After the convolution and pooling processes, the feature patterns, $x_1$ and $x_2$ (middle part of Figure 7), indicated the different dynamic trajectories in amplitudes in the systole region (systolic peak) and the systolic peak to diastolic peak region (time delay), which showed the similarities or differences between Nor, LPAD, and HPAD.

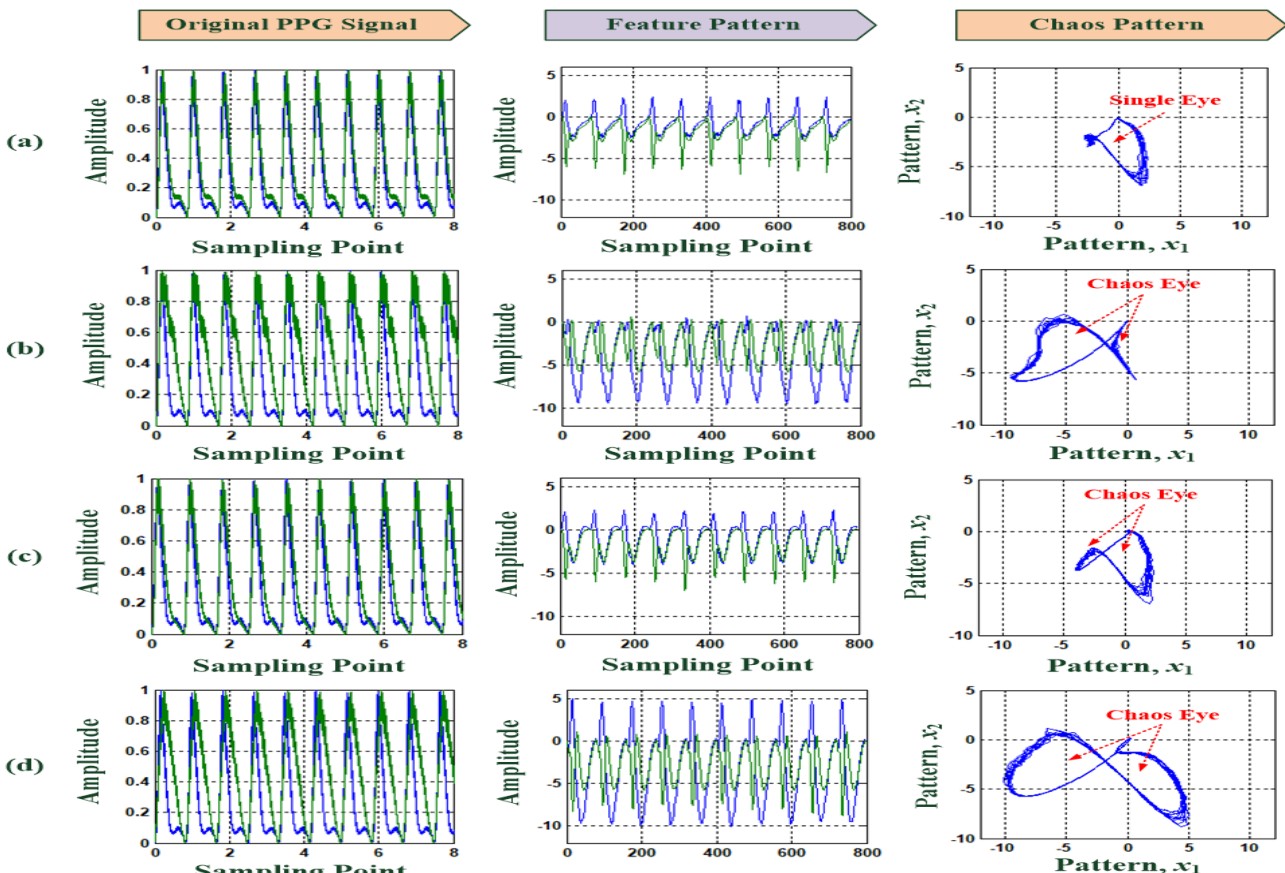

**Figure 7.** Original PPG signals (10 cycles), feature patterns (after the pooling processes), and chaos patterns for different PAD risk levels. (**a**) Nor: single eye in the chaos pattern, (**b**,**c**) LPAD cases, (**d**) HPAD case.

In the chaos patterns for the three risk levels, the $x_1$ and $x_2$ feature patterns (the dimension of each feature pattern is $1 \times 200$) can be utilized to plot the dynamic error scatter diagram, as $x_1$ and $x_2$ are in the horizontal and vertical axes, respectively, where the blue lines denote different dynamic trajectories. Meanwhile, the waveform's amplitude and time delay were different, and the chaotic trajectories indicated different distribution regions that gradually separated in the third and fourth quadrants and exhibited two chaos eyes (shown as red dashed lines in Figure 7). Hence, the chaos patterns showed that the differences in chaotic dynamics were the effect of the differences in PPG morphology, which could be used to separate Nor from LPAD/HPAD. Thus, the proposed feature patterns could be validated for UEVD screening.

### 3.2. D CNN-Based Classifier Training and Testing

With the PPG measurement data, we could extract a total of 160 feature patterns from 40 subjects, including 44 normal patterns and 116 abnormal patterns (44 LPADs and 72 HPADs). In the training stage, 10-fold, $K_f = 10$, cross-validation was performed by interchanging trained and untrained feature patterns. We could randomly select 40 trained feature patterns to train the multilayer classifier. Hence, using 40 pairs of input–output patterns, we obtained 200 input nodes, 40 pattern nodes, 4 summation nodes, and 3 output nodes in the fully connected network topology (network topology = $200 - 40 - 4 - 3$), as shown in Figure 5.

Then, setting the convergent condition, that is, tolerance value $\varepsilon \leq 10^{-2}$, and the initial condition, that is, $\sigma_0 = 1.0000$, the gradient descent method was used to minimize the mean squared error (MSE) using the iteration computation to tune the optimal smoothing parameters in the pattern nodes. Figure 8a,b show the training history curves for the training classifier, as optimal parameters versus iteration numbers and MSEs versus iteration numbers, respectively. With the different learning rates, $\eta = 0.1$–$0.5$, the gradient descent method required <25 iterative computations to reach the specific convergent condition, as shown in Figure 8b. The average optimal parameter, $\sigma_{opt} = 0.3360$, could be utilized to increase the classification accuracy. The iteration processes required an average of 0.0960 s of CPU time to determine the optimal parameter.

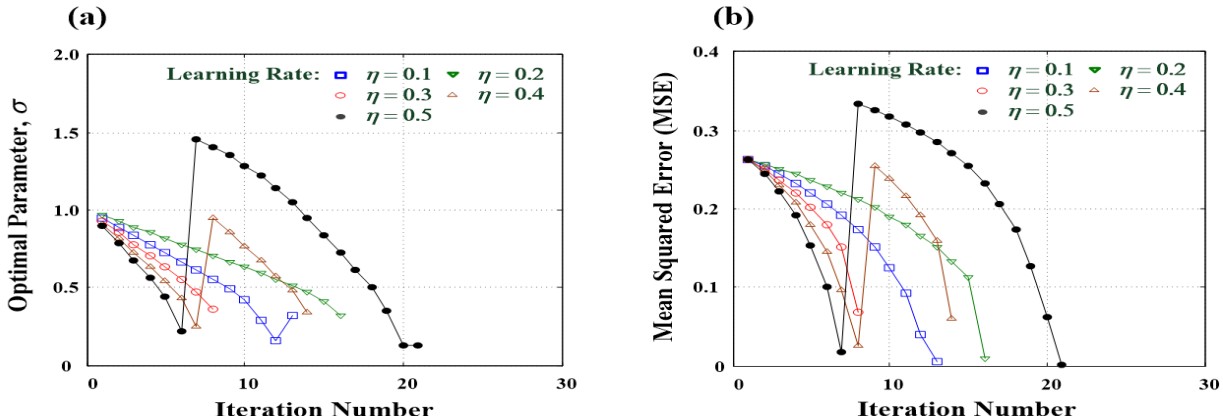

**Figure 8.** Training history curves for training the multilayer classifier. (**a**) Optimal parameters versus iteration numbers, (**b**) mean squared errors versus iteration numbers.

Moreover, with a high learning rate, $\eta \geq 0.4$, the convergent condition for determining the optimal parameter could be rapidly achieved. However, its training history curves indicated unstable convergent performances or divergent conditions in the training stage, as denoted by the symbols -•- and -Δ- shown in Figure 8. By contrast, with a low learning rate, $\eta < 0.4$, the learning algorithm could guarantee that the convergent condition will be achieved. Moreover, its training history curves indicated a monotonous decrease, as denoted by the symbols -O- and -∇- shown in Figure 8. The classification accuracy in the learning stage was 100.00%. For untrained feature patterns, with the 10-fold cross-

validation, we randomly select testing patterns from the tested 120 datasets to validate the performance of the proposed 1D CNN-based classifier. Table 3 shows the experimental results of the 10-fold cross-validation, with a mean recall of 97.92%, a mean precision of 94.48% for screening abnormality (i.e., true positive (TP), including both LPAD and HPAD), a mean accuracy of 94.50% for the correctly identified risk levels, and a mean $F_1$ score of 0.9615 for the evaluated classifier performances. The index of the $F_1$ score is the weighted average of recall and precision, which considers both false positive (FP) and false negative (FN). The $F_1$ score is more useful than accuracy in evaluating classifier performance. We obtained an $F_1$ score of 0.9615, which indicates approximately 96% accuracy in separating normal patterns from abnormal patterns. Recall as the sensitivity index correctly identifies TP observations from all observations in the actual classification task. Hence, the recall indicator was the true accuracy of TP, and the precision indicator was the standard for predicting TP. Both indicators were greater than 80%. The recall indicator is also called the positive predictive value (PPV). In general, the PPV index was >80%, indicating that the classifier had a good predictive performance. Furthermore, Youden's index (YI) used both sensitivity and specificity to quantify the number of FPs and FNs, and its value ranged from 0 to 1. Hence, in this study, the average of YI = 0.8473 ≥ 0.5 was obtained. Hence, the larger the YI is, the better the performance of our proposed 1D CNN-based classifier in UEVD screening and the greater its authenticity for evaluating the probability of an informed decision. Therefore, our proposed 1D CNN-based classifier had promising results for UEVD screening.

**Table 3.** Experimental results of tenfold cross-validation for the proposed 1D CNN-based classifier.

| Cross-Validation | Trained Patterns | Untrained Patterns | Recall (%) | Precision (%) | Accuracy (%) | F1 Score | Youdens Index |
|---|---|---|---|---|---|---|---|
| 1 | | | 100.00 (TP: 27, FN: 0) | 93.10 (TP: 27, FP: 2) | 95.00 (2 failures) | 0.9643 | 0.8462 |
| 2 | | | 100.00 (TP: 28, FN: 0) | 96.55 (TP: 28, FP: 1) | 97.50 (1 failures) | 0.9825 | 0.9167 |
| 3 | | | 96.43 (TP: 27, FN: 1) | 93.10 (TP: 27, FP: 2) | 92.50 (3 failures) | 0.9474 | 0.7976 |
| 4 | Random Selection Normality: 11 patterns Abnormality: 29 patterns Total of Trained Pattern: 40 | Random Selection Normality: 11 patterns Abnormality: 29 patterns Total of Trained Pattern: 40 | 100.00 (TP: 27, FN: 0) | 93.10 (TP: 27, FP: 2) | 95.00 (2 failures) | 0.9643 | 0.8462 |
| 5 | | | 100.00 (TP: 28, FN: 0) | 96.55 (TP: 28, FP: 1) | 97.50 (1 failures) | 0.9825 | 0.9167 |
| 6 | | | 93.33 (TP: 28, FN: 2) | 96.55 (TP: 28, FP: 1) | 92.50 (3 failures) | 0.9492 | 0.8333 |
| 7 | | | 96.43 (TP: 27, FN: 1) | 93.10 (TP: 27, FP: 2) | 92.50 (3 failures) | 0.9474 | 0.7976 |
| 8 | | | 96.43 (TP: 27, FN: 0) | 93.10 (TP: 27, FP: 2) | 95.00 (2 failures) | 0.9643 | 0.8462 |
| 9 | | | 96.43 (TP: 27, FN: 1) | 93.10 (TP: 27, FP: 2) | 92.50 (3 failures) | 0.9474 | 0.7976 |
| 10 | | | 96.55 (TP: 28, FN: 1) | 96.55 (TP: 28, FP: 1) | 95.00 (2 failures) | 0.9655 | 0.8746 |
| Average (%) | | | 97.92 | 94.48 | 94.50 | 0.9615 | 0.8473 |

Note: (1) TP: true positive, (2) FN: false negative, (3) FP: false positive.

## 4. Conclusions

We have developed a 1D CNN-based classifier for UEVD screening, including Nor, LPAD, and HPAD. The 1D CNN consisted of a D–H-based quantizer, two 1D digital convolution operators, two 1D subsampling processes, and a fully connected network (as shown in Figure 5). The D–H-based quantizer was employed to extract the self-synchronization dynamic errors between normal and abnormal PPG signals, which could indicate the differences in PPG morphology, such as the region of systolic peak and the time interval

between systolic and diastolic peaks. Then, the self-synchronization dynamic errors were used to separate normal patterns from abnormal patterns, including Nor and LPAD/HPAD. The 1D digital convolution operator was used to enhance the feature pattern and remove the unwanted noises, and the dimension of the feature pattern could be reduced by the 1D subsampling process from $1 \times 1000$ to $1 \times 100$ (as shown in Figures 6 and 7). Finally, the feature patterns were fed into the multilayer classifier, the UEVD risk level could be identified in the PPG stream data, and the screening tasks were performed. With the 10-fold cross-validation, at each cross-validation, given 40 randomly selected untrained feature patterns, the proposed classifier had an average recall of 97.92%, average precision of 94.48%, average accuracy of 94.50%, and average $F_1$ score of 0.9615 for screening abnormalities (i.e., LPAD and HPAD). The average of YI = $0.8473 \geq 0.5$ could meet empirical benchmarks for being administered for diagnostic purposes and provide a measure of the capability of a diagnostic test to balance sensitivity and specificity. Experimental results of 10-fold cross-validation indicated the computational efficiency and promising screening capability for a limited training dataset to predict the untrained dataset. The experimental results showed that the proposed 1D CNN-based classifier had promising performances in 1D raw PPG signal processing in the time domain and could rapidly extract and enhance features for further classification tasks with the D–H-based quantizer and 1D convolution processes and reduce the feature dimension in the 1D subsampling processes, resulting in low computational operations for real-time signal screening applications. Hence, the proposed wrist-based PPG wearable tool and 1D multilayer classifier could be used for early detection in a nonmedical environment and continuous home healthcare monitoring. Cooperating with the image examinations, such as vascular ultrasound, catheter angiography, and computed tomography angiography could create images to view the areas of restricted blood flow and then to locate blockages. These manners could provide a reliable examination to evaluate the vascular condition for further surgery or interventional therapy recommendations. In addition, in real-world applications, PPG measurements with annotations were continuously obtained, the new training dataset could be added to the current database for further trained the classifier. The proposed 1D CNN could rapidly retain a new pattern mechanism by adding a new trained dataset. Hence, the 1D multilayer classifier could keep its intended medical purpose in real-world application and could also be used as a computer-aided decision-making tool and software in a medical device tool.

**Author Contributions:** Conceptualization: P.-Y.C., C.-C.P. and C.-M.L.; analysis and materials: P.-Y.C., J.-X.W., C.-H.L. and Z.-L.S; data analysis: P.-Y.C., J.-X.W., C.-H.L. and Z.-L.S.; writing—original draft preparation: P.-Y.C., C.-H.L. and N.-S.P.; writing—review and editing: P.-Y.C., J.-X.W., C.-H.L. and N.-S.P.; supervision: P.-Y.C., C.-H.L. and N.-S.P.; funding acquisition: P.-Y.C., J.-X.W. and C.-H.L. All authors have read and agreed to the published version of the manuscript.

**Funding:** This work was supported by the Ministry of Science and Technology, Taiwan, under contract number: MOST 108-2218-E-167 -00-MY2, MOST 108-2221-E-167-005-MY2, and MOST 110-2635- E- 167-033, duration: 1 August 2019–31 July 2022.

**Institutional Review Board Statement:** All subjects gave their informed consent for inclusion before they participated in the study. The study was conducted in accordance with the Declaration of Helsinki, and the enrolled data was also approved by the hospital research ethics committee and the Institutional Review Board (IRB).

**Informed Consent Statement:** Not applicable.

**Data Availability Statement:** The data presented in this study are available in the article.

**Conflicts of Interest:** The authors declare no conflict of interest.

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
