# Peer review of "Photoplethysmography Analysis with Duffing–Holmes Self-Synchronization Dynamic Errors and 1D CNN-Based Classifier for Upper Extremity Vascular Disease Screening"

_processes, doi:10.3390/pr9112093_

Round 1

Reviewer 1 Report

this is an interesting study in which the authors attempt to close the gaps arising from 2D CNNs, so a 1D CNN with a time-domain feature extraction layer (i.e., D-H-based quantizer), 1D convolution processes, 1D subsampling processes, are designed for automatic Upper extremity vascular disease screening.

(1) Can the authors quote some statistics or share  more background information about this disease? More comparison studies also need to be included.

(2) Line 105-106: This adaptive learning model can also be deployed to adjust the optimal network parameter with feeding training patterns using the gradient descent method or particle swarm optimization algorithm [30, 37–39]---> I will like the authors to elaborate on this further.

(3) Figures --- as the figures formatting look very different throughout, i will like to check if they are all self-generated by the authors? If not, there is a need to acknowledge/state the reference from which the figure is obtained.

(4) why is Line 233-238 in blue? its looks weird. 

(5) the manuscript generally reads well, but the main limitation of the study is the small sample size. Authors will need to explicilty state this as the limitation of their study. Do the authors intend to validate their model with a larger dateset? Is it possible to do so to enahance their study?

(6) I was hoping to read more about the future work but could not find it. Authors should discuss their futue work. 

The manuscript can be accepted after the above concerns have been addressed.

Author Response

For Reviewer: 1#

This is an interesting study in which the authors attempt to close the gaps arising from 2D CNNs, so a 1D CNN with a time-domain feature extraction layer (i.e., D-H-based quantizer), 1D convolution processes, 1D subsampling processes, are designed for automatic Upper extremity vascular disease screening.

Response: Thank you for review’s comments. The point-to-point responses to all the referees are shown below.

(1) Can the authors quote some statistics or share more background information about this disease? More comparison studies also need to be included.

Response: Thank you for reminding us. Some sentences have been added in Introduction, Page#1.

(2) Line 105-106: This adaptive learning model can also be deployed to adjust the optimal network parameter with feeding training patterns using the gradient descent method or particle swarm optimization algorithm [30, 37–39]---> I will like the authors to elaborate on this further.

Response: Thank you for reminding us. Some sentences have been added in Introduction, Page#3.

(3) Figures --- as the figures formatting look very different throughout, i will like to check if they are all self-generated by the authors? If not, there is a need to acknowledge/state the reference from which the figure is obtained.

Response: Thank you for reminding us. Yes, all figures are self-generated by us.

(4) why is Line 233-238 in blue? its looks weird. 

Response: Thank you for reminding us. The blue highlight has been deleted.

(5) the manuscript generally reads well, but the main limitation of the study is the small sample size. Authors will need to explicilty state this as the limitation of their study. Do the authors intend to validate their model with a larger dataset? Is it possible to do so to enahance their study?

Response: Thank you for reminding us. Some sentences have been added in Conclusion, Page#11.

(6) I was hoping to read more about the future work but could not find it. Authors should discuss their future work.

Response: Thank you for reminding us. Some sentences have been added in Conclusion, Page#11.

Reviewer 2 Report

Dear Authors,

It was a pleasure for me to review such an interesting article on a relevant topic. The results obtained can be useful in the field of personalized medicine and health care. However, there are several reasons that prevent me from recommending the manuscript for publication in its current form.

1. It is not entirely clear from the manuscript where the normal and defective signals of the photoplethysmogram are supposed to be obtained from, so that these signals are in-phase?

2. Why use a chaotic system and neural networks if the signals to be classified are different in the time domain? Why can't the Fourier transform be used? It is required to compare the quality of recognition of two different signals using the proposed approach and the Fourier transform.

3. Why is the Duffing Oscillator chosen as a chaotic system? Can it be replaced with a discrete map to increase performance?

4. Simulation details should be added. In particular, it is necessary to indicate which numerical method was used to solve the ODE system. This is important because some integration methods introduce additional sampling effects that distort the study results.

However, my overall impression is positive, so I recommend re-reviewing the manuscript after major revisions.

Author Response

For Reviewer: 2#

It was a pleasure for me to review such an interesting article on a relevant topic. The results obtained can be useful in the field of personalized medicine and health care. However, there are several reasons that prevent me from recommending the manuscript for publication in its current form.

1. It is not entirely clear from the manuscript where the normal and defective signals of the photoplethysmogram are supposed to be obtained from, so that these signals are in-phase?

Response: Thank you for reminding us. Some sentences have been added in Introduction and Section 2.1., Pages#2, #3, and #4.

2. Why use a chaotic system and neural networks if the signals to be classified are different in the time domain? Why can't the Fourier transform be used? It is required to compare the quality of recognition of two different signals using the proposed approach and the Fourier transform.

Response: Thank you for reminding us. Some sentences have been added in Introduction, Page#2.

3. Why is the Duffing Oscillator chosen as a chaotic system? Can it be replaced with a discrete map to increase performance?

Response: Thank you for reminding us. Some sentences have been added in Section 2.2., Page #

4. Simulation details should be added. In particular, it is necessary to indicate which numerical method was used to solve the ODE system. This is important because some integration methods introduce additional sampling effects that distort the study results.

Response: Thank you for reminding us. Some sentences have been added in Introduction, Page #3.

Round 2

Reviewer 1 Report

The revised manuscript can now be accepted.

Reviewer 2 Report

Dear Authors,

Thank you for taking my recommendations into account. I recommend the article for publication.